# Optimised Workflows for Profiling the Metabolic Fluxes in Suspension vs. Adherent Cancer Cells via Seahorse Technology

**DOI:** 10.3390/ijms26010154

**Published:** 2024-12-27

**Authors:** Eugenia Giglio, Martina Giuseffi, Simona Picerno, Marzia Sichetti, Marisabel Mecca

**Affiliations:** Laboratory of Preclinical and Translational Research, Centro di Riferimento Oncologico della Basilicata (IRCCS-CROB), Rionero in Vulture, 85028 Potenza, Italy; eugenia.giglio@crob.it (E.G.); martina.giuseffi@crob.it (M.G.); simona.picerno@crob.it (S.P.); marzia.sichetti@crob.it (M.S.)

**Keywords:** cancer metabolism, bioenergetics, glycolysis, oxidative phosphorylation, ATP rate, suspension cells, adherent cells, Seahorse technology

## Abstract

Oxidative phosphorylation and glycolysis are the main ATP-generating pathways in cell metabolism. The balance between these two pathways is frequently altered to carry out cell-specific activities in response to stimuli involving activation, proliferation, or differentiation. Despite being a useful tool for researching metabolic profiles in real time in relatively small numbers of cancer cells, the main Agilent Seahorse XF Pro Analyzer (Agilent Technologies, Santa Clara, CA, USA) guideline is currently not fully detailed in the distinction between suspensions vs. adherent cancer cells. This article provides step-by-step protocols for profiling metabolic fluxes in suspension vs. adherent cancer cells via Seahorse technology, including adjustments for normalisation of data on the basis of the number of viable cells or the total protein content. Owing to the adaptations of plates, reagents, cell count, and protein quantification, it is possible to (i) analyse both adherent and suspension cells with a single instrument; (ii) conduct all experiments in 96-well plates, thus using fewer cells, media, and reagents; (iii) determine the effect of a drug or compound directly on cell metabolism; (iv) normalise data on the basis of the number of viable cells or the total protein content via a spectrophotometer; and (v) achieve notable savings in cost and time.

## 1. Introduction

One of the major hallmarks of tumorigenesis, metastasis, and drug resistance is abnormal cell metabolism, which includes aerobic glycolysis and anabolic respiration [1]. Glycolysis and oxidative phosphorylation are the two primary pathways for adenosine triphosphate (ATP) production. Mitochondria are the major cellular source of ATP in nonproliferating cells and are relatively minor contributors to ATP production in several cancers [2,3,4]. Indeed, the mitochondrial respiratory chain converts the chemical energy of pyruvate and other carbon-rich substrates (such as fatty acids and amino acids) into potential energy, which drives ATP synthesis [5,6].

Therefore, mitochondria have emerged as potential targets for anticancer therapy since they are structurally and functionally different from their noncancerous counterparts [7].

One of these differences in the metabolism of cancer cells was observed in the early 20th century by Warburg, who reported that tumours produce more lactic acid than differentiated cells even in the presence of oxygen. As a result of this contribution, the tumour metabolic phenotype was named the Warburg effect [2,8,9].

This effect is characterised by increased glycolysis and the inhibition of mitochondrial metabolism even in the presence of physiological levels of oxygen.

Since these initial studies, there have been additional interesting studies carried out over the years to support this theory. Rios et al. [10] highlighted that adherent and suspension cells have different metabolic profiles; specifically, the non-adherent state has higher mitochondrial activity. According to another study by Vannocci et al. [11], adherent cells seemed to be more dependent on lactic fermentation and glycolytic turnover. Whereas suspended cells show a decrease in glycolytic reserve, which measures a cell’s ability to compensate for the increase in energy demand through glycolysis.

A recent methodology for accurately measuring cellular metabolism has advanced the understanding of the Warburg effect and its importance in facilitating anabolic processes linked to cancer cell growth and proliferation [12].

An essential first step in investigating the metabolic profile of tumours is the use of in vitro cancer models.

The use of Oroboros O2K and Agilent Seahorse technologies for studying metabolism is expanding in this field. Oroboros O2K (Oroboros Instruments, Innsbruck, Austria) is a high-resolution respirometer (HRR) for measuring rates of mitochondrial respiration. However, the Oroboros O2K has some limitations. It only has two chambers, which makes high-throughput testing impractical without multiple O2K units. Additionally, the cleaning protocols after each experiment are critical and procedure-specific, requiring up to three hours [13]. Meanwhile, the Agilent Seahorse Extracellular Flux (XF) Pro Analyzer (Agilent Technologies, Santa Clara, CA, USA) is a fully integrated 96-well instrument that analyses cellular metabolism in real time directly into a cell culture plate. It measures the flux of oxygen (oxygen consumption rate, OCR) and protons (extracellular acidification rate, ECAR) of mitochondrial respiration and glycolysis, respectively, as well as the ATP production rate. OCR data measure mitochondrial oxidative phosphorylation because oxygen is the final electron acceptor during aerobic metabolism. Instead, the ECAR essentially assesses the rate of glycolytic metabolism by measuring the amount of pyruvate produced and its subsequent conversion to lactate. This conversion takes place under anaerobic conditions to restore NAD^+^ and enable the continuous synthesis of glycolytic ATP [14]. The Seahorse XF Pro Analyzer can also monitor fluctuations in the OCR and ECAR following treatment with molecules or drugs of interest, employing either custom-assembled or preformulated drug combinations available in each specific kit. This approach has provided a significant advantage since methods to evaluate the potential of a new drug to induce mitochondrial dysfunction are currently needed in drug development [15].

This technology has been used for metabolic phenotyping of cancer cells under both perturbed and standard conditions [16] to discover metabolic switches that confer malignant characteristics (e.g., metastasis) [17] and to investigate metabolic heterogeneity in cancer by identifying cellular subpopulations that harbour various metabolic profiles (e.g., cancer stem cells) [18,19].

Agilent Seahorse XF Pro Analyzer measures the OCR and ECAR in real time at intervals of approximately 5–8 min. Typically, baseline (basal) OCR (pmol/min) and ECAR (mpH/min) rates are measured three or four times before the addition of inhibitors, stimulants, substrates, or compounds. Seahorse XF measurements are non-invasive, allowing for repeated metabolic assessments of the same cell in multiple assay wells. To maintain normal cell physiology, a temperature control system maintains the cellular environment at 37 °C. The total assay time is typically 60–90 min, and upon completion of an XF assay, other types of biological assays can be performed on the same plate.

The Seahorse XF Pro Analyzer uses a sensor cartridge with two fluorophores for measuring oxygen and pH. In particular, directly into microplates, the sensor probes generate a transitory microchamber of 200 microns at the bottom of the well. The variations in dissolved oxygen and free proton concentrations caused by cellular oxygen intake (respiration) and proton excretion (glycolysis) are subsequently evaluated. The Seahorse XF Pro Analyzer estimates the OCR and ECAR after these measurements are taken for 2–5 min. Once a measurement is completed, the probes are raised, the cell values are returned to baseline values, and additional measurements are allowed.

Seahorse Wave 2.4 XF-96 software interpolates independently measured values of ATP production by glycolysis and the mitochondrial respiratory chain via oligomycin, an inhibitor of respiratory chain complex V, and rotenone/antimycin A, complex I and III inhibitors, respectively [14] (Figure 1).

Several kits can be used depending on the type of study on cellular energy metabolism. The Seahorse XF Real-Time ATP Rate Assay Kit allows the quantification of the rate of ATP production from glycolysis and oxidative phosphorylation simultaneously via oligomycin and rotenone/antimycin A.

The glycolytic ATP production rate, the rate of ATP production in the glycolytic pathway, is equal to the glycolytic proton efflux rate (glycoPER) (Equation (1)).
(1)$$glycoATP\;Production\;Rate\left(\frac{pmol\;ATP}{min}\right)=glycoPER\left(\frac{pmol\;H^{+}}{min}\right)$$

Instead, the rate of oxygen consumption associated with ATP synthesis during oxidative phosphorylation can be determined as the OCR that is inhibited by oligomycin (OCR_Oligo_):(2)$${OCR}_{ATP}\left(\frac{pmol\;O_{2}}{min}\right)=OCR\left(\frac{pmol\;O_{2}}{min}\right)-{OCR}_{Oligo}\left(\frac{pmol\;O_{2}}{min}\right)$$

Overall, through the following equation (Equation (3)), Seahorse XF software calculates the transformation of OCR_ATP_ to the mitochondrial ATP production rate:(3)$$mitoATP\;Production\;Rate\left(\frac{pmol\;ATP}{min}\right)={OCR}_{ATP}\left(\frac{pmol\;O_{2}}{min}\right)\times 2\left(\frac{pmol\;O}{pmol\;O_{2}}\right)\times \frac{P}{O}\left(\frac{pmol\;ATP}{pmol\;O}\right),$$
where the P/O ratio refers to the number of molecules of ATP generated per atom of oxygen reduced during cellular respiration. It is a gauge of the efficiency of oxidative phosphorylation in generating ATP.

To conclude, the total cellular ATP production rate is calculated as the sum of the glycolytic and mitochondrial ATP production rates, according to Equation (4):(4)$$ATP\;Production\;Rate\left(\frac{pmol\;ATP}{min}\right)=glycoATP\;Production\;Rate\left(\frac{pmol\;ATP}{min}\right)+mitoATP\;Production\;Rate\left(\frac{pmol\;ATP}{min}\right)$$

As a starting point for the investigation and evaluation of the metabolic profile of tumour cells, this protocol proposes two detailed ATP rate assay procedures to compare the application of Seahorse XF technology between suspension and adherent cancer cells. This study aims to elucidate and highlight the differences between these two protocols in a single paper and to facilitate the study of cellular metabolism via Seahorse technologies.

## 2. Materials and Methods

### 2.1. ATP Rate Assay Protocol for Suspended Cells

Required materials:Define cultured (or freshly isolated) suspension cells;37 °C non-CO_2_ incubator;Incubator set to 37 °C with CO_2_;Centrifuge with a microplate rotor;Agilent Seahorse Extracellular Flux (XF) Pro Analyzer (Agilent Technologies, Santa Clara, CA, USA);Seahorse XFe96/XF Pro PDL FluxPak (#103798-100, Agilent Technologies, Santa Clara, CA, USA, including PDL cell culture microplates, Sensor cartridges and Seahorse XF Calibrant Solution);Seahorse XF RPMI medium (#103576-100) or Seahorse XF DMEM medium (#103575-100);Seahorse XF 1.0 M glucose solution (#103577-100);Seahorse XF 100 mM pyruvate solution (#103578-100);Seahorse XF 200 mM glutamine solution (#103579-100);Seahorse XF Real-Time ATP Rate Assay Kit (#103592-100);Additional reagents and equipment for counting cells.

NOTE: All procedures should be carried out at room temperature in a sterile environment unless otherwise specified.

Importantly, all protocols require the use of the newest instrument (XF Pro), but they may also be carried out on older models (XF machines) with the same materials (XF FluxPak). Furthermore, for run accuracy, the use of at least four replicate wells per group is recommended. However, obtaining consistent outcomes from separate tests is essential for confirming findings.

This step-by-step protocol consists of the following steps: sensor cartridge hydration, culture of suspension cells the day prior to the XF test, and reseeding of the cells and assay on the day of the test (Figure 2). All steps have been optimised for examining metabolism in primary or immortalised suspension cancer cells.

#### 2.1.1. Day Prior to XF Assay

The Seahorse XF Real-Time ATP rate assay test kit allows the characterisation of the cell type of interest in a single assay.

Suspension cells require high seeding density, from 5 × 10^4^ to 2 × 10^5^ (50–90% confluency) cells per well, depending on cell type.

Therefore, the day before the analysis:Seed a greater number of cells needs to be seeded in culture plates or flasks in order to assess changes in the metabolic profile following specific experimental conditions (e.g., drug treatments, silencing, starvation, etc.) (Figure 2);

NOTE: this ensures the minimum cell density required on the day of the assay.

Switch on the Seahorse XF Pro Analyzer and allow temperature to stabilize;Warm at least 20 mL of XF calibrant solution in a non-CO_2_ 37 °C incubator;Fill each well of the utility plate of the sensor cartridge with 200 μL of calibrant solution, place the XF Hydrobooster on top of the utility plate, and lower the sensor cartridge through the openings on the XF Hydrobooster plate into the utility plate, submerging the sensors in XF calibrant (Figure 3a);Put the microplate (with lid) back up to hydrate all probes into the non-CO_2_ 37 °C incubator overnight.

TECHNICAL TIPS: Ensure the sensor cartridge is not exposed to light, the cartridge should be left in the plastic shipping case until the overnight hydration.

#### 2.1.2. Day of XF Assay

Prepare the assay medium (50 mL of XF-RPMI or DMEM) by supplementing 10 mM of XF-glucose, 1 mM of XF-pyruvate, and 2 mM of XF-glutamine under sterile conditions and store it at 37 °C (Figure 2);Collect the cells from the culture plate/flask, and centrifuge them at 400 g for 5 min. After, wash the cells with XF medium twice and re-suspend the pellet in 210 μL of XF medium;

TECHNICAL TIPS: XF medium is light sensitive, ensure to protect it from light.

TECHNICAL TIPS: ensure do not leave any growth medium containing phenol red, as it may interfere with the assay.

3.Count the cells and adjust the concentration to 1 × 10^6^ cells/mL with XF medium (Figure 2);4.Seed 50 μL of cell suspension per well in Seahorse XF Pro PDL Cell Culture Microplate; put only XF medium (no cells) in background correction wells (A1, A12, H1, H12) (Figure 4);

TECHNICAL TIPS: if wells A1, A12, H1, or H12 are loaded by mistake, use water or exposure to air to kill cells, and rinse.

5.Gently Centrifuge the plate with a microplate rotor (~200× *g* for 5 min) to immobilize cells into a monolayer on the well bottom and then incubate at 37 °C, in a non-CO_2_ incubator for 30 min;

TECHNICAL TIPS: ensure cells are uniformly and seeded in a monolayer configuration; cell clusters may cause poor cell adhesion and inaccurate measurements of OCR.

6.Monitor adherence using a microscope (Axio vert A1, Zeiss, Jena, Germany);7.Slowly add 130 μL of XF medium to each well, placing the pipet tip against the side at the higher end in the well, and monitor adherence again using a microscope;8.Incubate cell plate at 37 °C in a non-CO_2_ incubator for 30 min;9.Prepare the stock solutions: resuspend the oligomycin and rotenon/antimycin A powders in 420 μL and 540 μL XF-media, respectively. Subsequently, vortex to ensure full resuspension of compounds;10.Prepare the working solutions: add 300 μL of stock solution to 2.7 mL of XF Assay Media per compound;

TECHNICAL TIPS: all regents and supplies are light sensitive, ensure to protect them from light, and store them at 4 °C.

TECHNICAL TIPS: use compounds the same day they are reconstituted. Do not freeze and discard any remaining compound.

11.Remove the hydrated cartridge from the non-CO_2_ incubator, place the A/B loading guide flat on top of the assay cartridge, and load injection solutions (Figure 3b):➢Standard AssayPort (A) 20 μL oligomycin (final concentration 1.5 µM).Port (B) 22 μL rotenone/antimycin A (final concentration 0.5 µM).➢Induced Assay (To inject a test compound)Port (A) 20 μL experimental compound or vehicle control.Port (B) 20 μL oligomycin (final concentration 1.5 µM).Port (C) 22 μL rotenone/antimycin A (final concentration 0.5 µM).

Position the multichannel pipette at a 45° angle on the doors to load the components (Figure 3b). During the procedure, it is useful to not make bubbles inside the doors, because they may cause solutions to leak from the injection port. To avoid this, touch the bottom with the tips.

NOTE: if performing a different type of XF assay, consult the corresponding XF Kit User Guide for appropriate loading methods for injection solutions.

TECHNICAL TIPS: if the injection ports are loaded improperly to the bottom of the well, stick a pipet tip down into the well to push the reagent to the bottom, avoiding suction and creating bubbles.

12.Design experiment in Wave Pro software opening XF Real-Time ATP Rate Assay template. Make any necessary group modifications to the template for the specific assay and map design: injection strategy, pretreatments, assay media, and cell type (name and seeding density) (Figure 4).

13.Place the sensor cartridge (hydrated and loaded with compounds) and utility plate onto the tray when prompted, and run calibration (Figure 2);

ERRORS: typical error messages are the notification that some assay wells have not been assigned to a group on the Plate Map, some background wells have not been defined, or no injection ports have been included in the assay protocol. Prior to running the assay, correct any errors that are displayed.

14.Upon completing the Calibration, open the tray to eject the utility plate and load the cell plate to initiate equilibration (the sensor cartridge remains inside the Seahorse XF Pro Analyzer for this step);15.After completing equilibration, the assay will automatically begin acquiring baseline measurements and take about 2 h to complete;16.Export and analyse data using Agilent Seahorse Analytics software (https://seahorseanalytics.agilent.com, accessed on 5 December 2024); the data can be normalized based on the number of seeded cells in the Seahorse XF Pro PDL Cell Culture Microplate before the analysis (Figure 2).

### 2.2. ATP Rate Assay Protocol for Adherent Cells

Required materials:Define cultured (or freshly isolated) adherent cells;RIPA Lysis and Extraction Buffer (#89901);Halt Phosphatase Inhibitor Cocktail (#1861277);Halt Protease Inhibitor Cocktail (#1862209);Protein Assay Dye Concentrate (#500-0006);Spectrophotometer Microplate Reader (PerkinElmer, Waltham, MA, USA);37 °C non-CO_2_ incubator;Incubator set to 37 °C with CO_2_;Centrifuge;Agilent Seahorse Extracellular Flux (XF) Pro Analyzer (Agilent Technologies, Santa Clara, CA, USA);Seahorse XFe96/XF Pro M FluxPak (#103777-100, including TC-treated cell culture microplates, Sensor cartridges and Seahorse XF Calibrant Solution;Seahorse XF RPMI medium (#103576-100) or Seahorse XF DMEM medium (#103575-100);Seahorse XF 1.0 M glucose solution (#103577-100);Seahorse XF 100 mM pyruvate solution (#103578-100);Seahorse XF 200 mM glutamine solution (#103579-100);Seahorse XF Real-Time ATP Rate Assay Kit (#103592-100);Additional reagents and equipment for counting cells.

NOTE: All procedures should be carried out at room temperature in a sterile environment unless otherwise specified.

Importantly, all protocols require the use of the newest instrument (XF Pro), but they may also be carried out on older models (XF machines) with the same materials (XF FluxPak). Furthermore, for run accuracy, the use of at least four replicate wells per group is recommended. However, obtaining consistent outcomes from separate tests is essential for confirming findings.

This step-by-step protocol consists of the following steps: sensor cartridge hydration and seeding of adherent cells the day prior to the XF test, and calibration and assay on the day of the test (Figure 5). All steps have been optimised for examining metabolism in primary or immortalised adherent cancer cells.

#### 2.2.1. Day Prior to XF Assay

The Seahorse XF Real-Time ATP rate assay test kit allows the characterisation of the cell type of interest in a single assay.

Suspended cells require a high seeding density, ranging from 5 × 10^4^ to 2 × 10^5^ (50–90% confluency) cells per well, depending on the cell type.

Therefore, the day before the analysis:Seed 150 µL of an appropriate number of cells needs in Seahorse XF Pro M (TC-treated) Cell Culture Microplate in order to assess changes in the metabolic profile following specific experimental conditions (e.g., drug treatments, silencing, starvation, etc.); put only XF medium (no cells) in background correction wells (A1, A12, H1, H12) (Figure 4);

NOTE: this ensures the minimum cell density required on the day of the assay.

TECHNICAL TIPS: if wells A1, A12, H1, or H12 are loaded by mistake, use water or exposure to air to kill cells, and rinse.

To analyze the metabolic profile, without specific experimental conditions, the cells can be seeded on the day of the assay directly in Seahorse XF Pro M Cell Culture Microplate (Figure 5);Switch on the Seahorse XF Pro Analyzer and allow temperature to stabilize;Warm at least 20 mL of XF calibrant solution in a non-CO_2_ 37 °C incubator;Fill each well of the utility plate of the sensor cartridge with 200 μL of calibrant solution, place the XF Hydrobooster on top of the utility plate, and lower the sensor cartridge through the openings on the XF Hydrobooster plate into the utility plate, submerging the sensors in XF calibrant (Figure 3a);Put the microplate (with lid) back up to hydrate all probes into a non-CO_2_ 37 °C incubator overnight.

TECHNICAL TIPS: Ensure the sensor cartridge is not exposed to light; the cartridge should be left in the plastic shipping case until the overnight hydration.

#### 2.2.2. Day of XF Assay

Prepare the assay medium (50 mL of XF-RPMI or DMEM) by supplementing 10 mM of XF-glucose, 1 mM of XF-pyruvate, and 2 mM of XF-glutamine under sterile conditions and store it at 37 °C (Figure 5);Monitor cells under a microscope to ensure adherence;

TECHNICAL TIPS: ensure cells are uniformly and seeded in a monolayer configuration; cell clusters may cause poor cell adhesion and inaccurate measurements of OCR.

3.Remove 150 μL of cell culture growth media from the cell culture microplate and wash once with 150 μL of preheated assay media;

TECHNICAL TIPS: XF medium is light sensitive, ensure to protect it from light.

TECHNICAL TIPS: ensure do not leave any growth medium containing phenol red, as it may interfere with the assay.

4.Slowly add 180 μL of XF medium to each well, placing the pipette tip against the side at the higher end of the well, and monitor adherence again using a microscope (Figure 5);5.Incubate cell plate at 37 °C in a non-CO_2_ incubator for 45–60 min;6.Prepare the stock solutions: resuspend the oligomycin and rotenon/antimycin A powders in 420 μL and 540 μL XF-media, respectively. Subsequently, vortex to ensure full resuspension of compounds;7.Prepare the working solutions: add 300 μL of stock solution to 2.7 mL of XF Assay Media per compound;

TECHNICAL TIPS: all regents and supplies are light sensitive, ensure to protect them from light, and store them at 4 °C.

TECHNICAL TIPS: use compounds the same day they are reconstituted. Do not freeze and discard any remaining compound.

8.Remove the hydrated cartridge from the non-CO_2_ incubator, place the A/B loading guide flat on top of the assay cartridge, and load injection solutions (Figure 3b):➢Standard AssayPort (A) 20 μL oligomycin (final concentration 1.5 µM).Port (B) 22 μL rotenone/antimycin A (final concentration 0.5 µM).➢Induced Assay (To inject a test compound)Port (A) 20 μL experimental compound or vehicle control.Port (B) 20 μL oligomycin (final concentration 1.5 µM).Port (C) 22 μL rotenone/antimycin A (final concentration 0.5 µM).

Position the multichannel pipette at a 45° angle on the doors to load the components. During the procedure, it is useful to not make bubbles inside the doors, because they may cause solutions to leak from the injection port. To avoid this, touch the bottom with the tips.

NOTE: if performing a different type of XF assay, consult the corresponding XF Kit User Guide for appropriate loading methods for injection solutions.

TECHNICAL TIPS: if the injection ports are loaded improperly to the bottom of the well, stick a pipette tip down into the well to push the reagent to the bottom, avoiding suction and creating bubbles.

9.Design experiment in Wave Pro software opening XF Real-Time ATP Rate Assay template. Make any necessary group modifications to the template for the specific assay and map design: injection strategy, pretreatments, assay media, and cell type (name and seeding density) (Figure 4).10.Place the sensor cartridge (hydrated and loaded with compounds) and utility plate onto the tray when prompted, and run calibration (Figure 5);

ERRORS: typical error messages are the notification that some assay wells have not been assigned to a group on the Plate Map, some background wells have not been defined, or no injection ports have been included in the assay protocol. Prior to running the assay, correct any errors that are displayed.

11.Upon completing the Calibration, open the tray to eject the utility plate and load the cell plate to initiate equilibration (the sensor cartridge remains inside the Seahorse XF Pro Analyzer for this step);12.After completing equilibration, the assay will automatically begin acquiring baseline measurements and take about 2 h to complete;13.Export and analyse data using Agilent Seahorse Analytics software (Figure 5); data can be normalized to total protein content;14.Post-analysis, wash cells with 200 μL PBS per well and proceed with lysis or freeze the microplate at −80 °C to ensure subsequent cell lysis;15.Subsequently, add 10 μL of RIPA buffer [50 mM Tris, pH 7.4, 150 mM NaCl, 1% (*w*/*v*) NP-40, 0.5% (*w*/*v*) sodium deoxycholate, and 0.1% (*w*/*v*) sodium dodecyl sulphate] per well, centrifuge the plate at 400× *g* for 5 min;16.Using a multichannel pipette, transfer 5 μL of supernatant to a new 96-well plate and add 5 μL of water (1:1 dilution) per well;17.Add 200 μL of previously diluted 1:5 H_2_O Protein Assay Dye Concentrate and read absorbance at 595 nm using a Spectrophotometer Microplate Reader.

## 3. Discussion and Conclusions

In the 20th century, Otto Warburg reported that, compared with non-tumour cells, tumours exhibit alterations in glycolysis and mitochondrial metabolism with a consequent increase in lactic acid [8,9]. For this reason, the study of metabolism and cellular regulation has become increasingly important. Hence, targeting metabolic pathways to treat cancer has aroused great interest. The initial methods employed for analysing ATP consumption, such as mass spectrometry or biochemistry-based methods, were limited because these measurements were performed on cell lysates and could not capture changes in real time. Furthermore, experimental noise may obscure modest but significant changes [20]. To overcome this problem, in recent years, new technologies, such as extracellular flux assays, have been developed to characterise bioenergetic changes through simultaneous measurements. It provides multiple advantages: higher throughput, reduced sample sizes, and enhanced kinetic resolution (several minutes) [21]. In particular, the Seahorse XF Pro Analyzer enables the measurement of oxygen consumption rates (OCRs) and extracellular acidification rates (ECARs), facilitating the estimation of the relative balance between oxidative phosphorylation and glycolysis, and thereby addressing the limitations associated with alternative methodologies. The Seahorse XF Pro Analyzer has led to the development of various experimental protocols and methodologies to provide a 360° view of cellular metabolism. Several studies have focused mostly on the use of the Seahorse system to investigate adherent cell metabolism. In contrast, for suspended cells in the literature, very few articles have focused on a single cell type. For this reason, we developed a protocol that outlines two comprehensive ATP rate assay procedures to investigate and evaluate the metabolic profiles of tumour cells, comparing the application of Seahorse XF technology in suspension with that in adherent cancer cells. This study aims to facilitate the study of cellular metabolism with these technologies.

Owing to the adaptations of plates, counts, and total protein content, it is possible to analyse both adherent and suspension cells with a single instrument, contrary to what was proposed in previous studies. For example, Zdrazilova et al. [22] used two distinct instruments to investigate whether respiration differed between adherent and suspended cells: Agilent Seahorse XF24 for adherent cells and Oroboros O2k for suspended cells.

However, Oroboros O2K has some limitations. It only has two chambers, which makes high-throughput testing impractical without multiple O2K units. It requires daily calibrations prior to each experiment to correct for room air oxygen concentrations and for the background correction of flux variability at high oxygen concentrations. Additionally, the cleaning protocols after each experiment are critical and procedure-specific, requiring up to three hours [13].

The Seahorse XF Pro Analyzer instead has only a few limitations: it lacks sensitivity for small-scale cellular oxygen consumption and lactate production measurement and requires only fresh cells because freezing and thawing can damage mitochondrial membranes and uncouple the electron transport system, complicating OCR measurement.

Therefore, our protocol not only adapts both cell types to a single instrument but also allows all experiments to be conducted in 96-well plates rather than 24-well plates, as described in other works [12]. The use of 96-well plates allows for the analysis of more experimental conditions in triplicate; moreover, it also allows the use of fewer cells, media, reagents, compounds, etc.

In addition, Seahorse XF technology allows one to determine the effect of a drug or compound directly on cell metabolism in real time by injecting it during cartridge loading if it has an immediate effect; otherwise, it can be added to the cells one or two days before the test and then analysed.

Another strength of this protocol is the normalisation of data on the number of viable cells or the total protein content, which is mandatory for the analysis of the metabolic profile of cancer cells. Hence, in the protocol adapted for suspension cells, the count of viable cells performed immediately before the analysis allows for normalisation of the data to this reliable value, thus avoiding lysis and the consequent quantification of proteins. However, the main limitation of this protocol adapted for suspension cells is the mandatory centrifugation to make the cells adhere to the bottom of the wells (which could stress the cells). On the other hand, the viable cell count conducted immediately before seeding the cells in the PDL-coated 96-well plate on the day of the assay allows for data normalisation based on the cell numbers, thus avoiding the quantification of total proteins, as is performed for adherent cells. In particular, in the protocol adapted for adherent cells, lysis is performed via a multichannel pipette directly on the 96-well plate used for the analysis with a small amount of RIPA buffer (10 μL), with the advantage of not having to detach and count the cells. Moreover, by adding the Protein Assay Dye reagent, a simple, accurate colorimetric assay based on the Bradford method, it is possible to quickly and simultaneously estimate the total protein content via a spectrophotometer microplate reader. This leads to notable savings in cost and time, especially compared with protocols where cells are counted via a fluorescence imaging system [23].

## Figures and Tables

**Figure 1 ijms-26-00154-f001:**
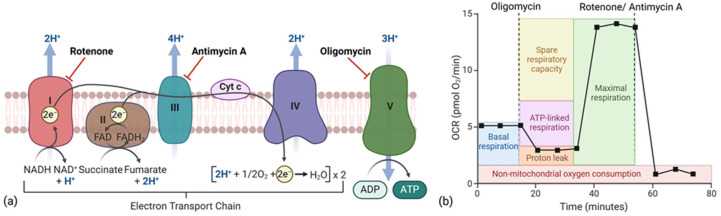
(**a**) Schematic illustration of the mechanism of rotenone, antimycin A, and oligomycin action at the mitochondrial respiratory chain site and (**b**) a corresponding OCR curve. Created in BioRender. Mecca, M. (2024) https://BioRender.com/p58j733 (accessed on 26 November 2024).

**Figure 2 ijms-26-00154-f002:**
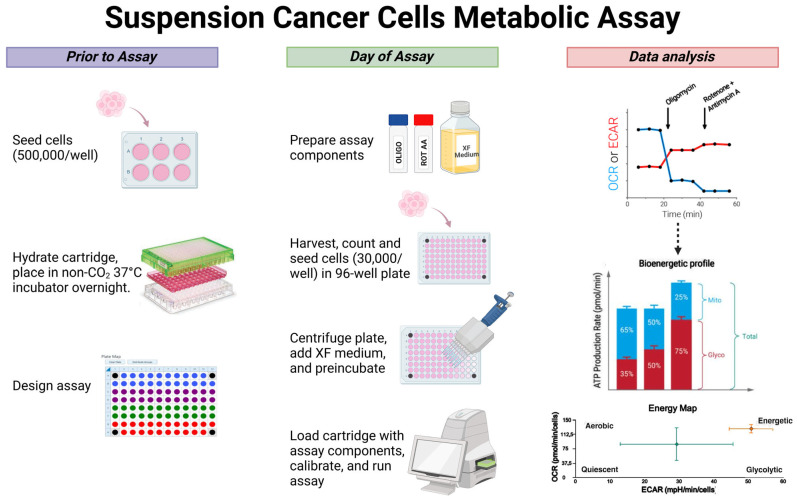
Experimental steps for profiling the metabolism of cancer cell suspensions: culture and seeding of cells in suspension, cartridge hydration, and design experiment in Seahorse Wave 2.4 XF-96 software on the day prior to assay; preparation of compounds and XF medium, counting and seeding of cells in PDL 96-well plate, centrifugation of the plate, cartridge calibration with assay components, and run assay on the day of assay; data normalization based on seeded cells number, and data analysis.Created in BioRender. Mecca, M. (2024) https://BioRender.com/n73r257 (accessed on 18 November 2024).

**Figure 3 ijms-26-00154-f003:**
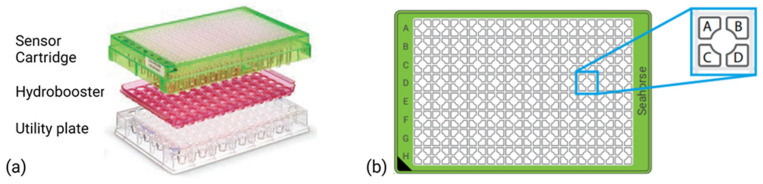
(**a**) Green sensor cartridge, pink hydrobooster and transparent utility plate suitable for calibration and hydration; (**b**) A–D injection ports for loading the Seahorse sensor cartridge. Created in BioRender. Mecca, M. (2024). https://BioRender.com/b90e604 (accessed on 26 November 2024).

**Figure 4 ijms-26-00154-f004:**
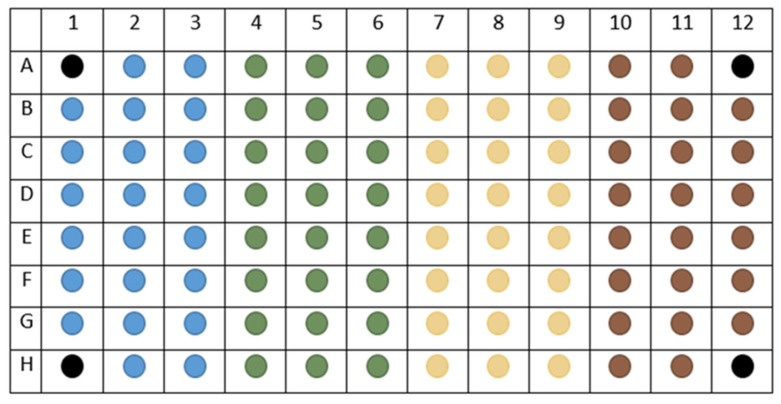
Example plate map. In the wells marked with black colour (A1, A12, H1, H12), there are no cells but only XF medium. Each experimental condition, performed in triplicate, corresponds to a different colour on the plate map.

**Figure 5 ijms-26-00154-f005:**
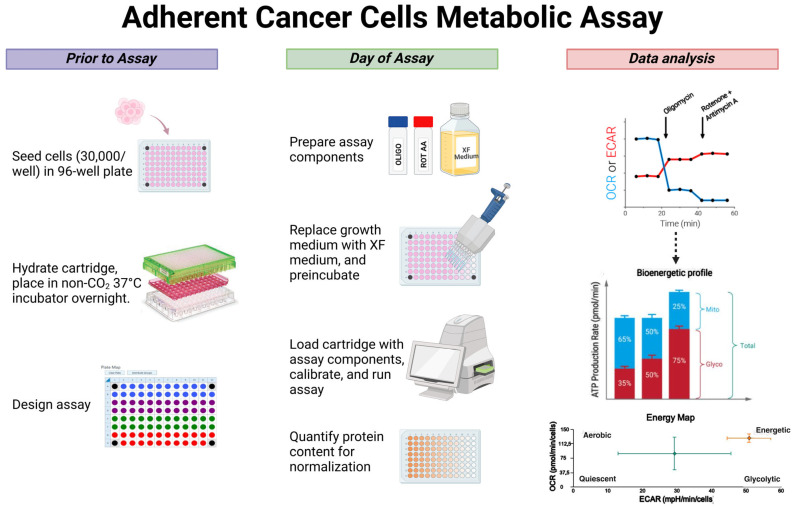
Experimental steps for profiling the metabolism of adherent cancer cells: seeding of cells in TC treated 96-well plate, cartridge hydration, and design experiment in Wave Pro software on the day prior to assay; preparation of compounds and XF medium, replacement of the culture medium with XF medium, pre-incubation of the plate, cartridge calibration with assay components, and run the assay on the day of assay; data normalization using total protein amount, and data analysis. Created in BioRender. Mecca, M. (2024) https://BioRender.com/s85b518 (accessed on 20 November 2024).

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
