# Peer review of "Optimised Workflows for Profiling the Metabolic Fluxes in Suspension vs. Adherent Cancer Cells via Seahorse Technology"

_ijms, 2024, doi:10.3390/ijms26010154_

Round 1

Reviewer 1 Report

Comments and Suggestions for Authors

The manuscript presents a detailed, well-organized protocol for profiling metabolic fluxes in both suspension and adherent cancer cells using Seahorse XF technology. The methods are clearly articulated, and the step-by-step instructions will be highly beneficial for researchers interested in cancer metabolism. However, there are a few areas that could be improved for clarity, precision, and comprehensiveness.

Abstract

Suggestion: Clarify the phrase "currently not completely optimised" by specifying what particular aspects are not optimized and how the study addresses these.

Contextual Depth: The introduction could benefit from more context on the clinical significance of distinguishing between suspension and adherent cell metabolism.

References: Ensure all claims (e.g., "recent methodologies have advanced understanding") are supported by citations.

Figure Legends: Ensure all figures have self-explanatory legends. For instance, Figure 2 could benefit from a brief description of the step-by-step workflow depicted.

Figure References: In the text, refer explicitly to each figure when relevant steps are discussed.

Minor Comments

Terminology Consistency: Ensure consistent use of terms like "XF Analyzer" vs. "Seahorse XF Pro Analyzer."

Typos:

Line 9: "metab-olism" should be corrected to "metabolism."

Line 15: "sus-pension" should be corrected to "suspension."

Abbreviations: Provide a brief list of abbreviations (e.g., OCR, ECAR) at the beginning or end for clarity.

Author Response

Reviewer 1

Comment 1: The manuscript presents a detailed, well-organized protocol for profiling metabolic fluxes in both suspension and adherent cancer cells using Seahorse XF technology. The methods are clearly articulated, and the step-by-step instructions will be highly beneficial for researchers interested in cancer metabolism. However, there are a few areas that could be improved for clarity, precision, and comprehensiveness.

Response 1: Thank you so much for your appreciation and your valuable advice and suggestions, which have greatly improved our Protocol;

Comment 2: Abstract

Suggestion: Clarify the phrase "currently not completely optimised" by specifying what particular aspects are not optimized and how the study addresses these.

Response 2: thank you for pointing this out, as suggested we have better specified on line 14 with: “the main Agilent Seahorse XF Pro Analyzer (Agilent Technologies, Santa Clara, CA, United States) guideline is currently not fully detailed in the distinction between suspensions versus adherent cancer cells”;

Comment 3: Contextual Depth: The introduction could benefit from more context on the clinical significance of distinguishing between suspension and adherent cell metabolism.

Response 3: thank you for pointing this out, as suggested, we have added on lines 44-56 “Since these initial studies, there have been additional interesting studies carried out over the years to support this theory. Rios et al. [10] have highlighted that adherent and suspension cells had different metabolic profiles; specifically, the non-adherent state had higher mitochondrial activity. According to another study by Vannocci et al. [11], adherent cells seemed to be more dependent on lactic fermentation and glycolytic turnover. Where-as suspended cells show a decrease in glycolytic reserve, which measures a cell’s ability to compensate for the increase in energy demand through glycolysis”, with the two relative references;

Comment 4: References: Ensure all claims (e.g., "recent methodologies have advanced understanding") are supported by citations.

Response 4: thank you, we have better specified the claim lines 57-59 as suggested, and we have added more references;

Comment 5: Figure Legends: Ensure all figures have self-explanatory legends. For instance, Figure 2 could benefit from a brief description of the step-by-step workflow depicted.

Response 5: thank you for pointing this out, as suggested we have added a brief description of the step-by-step workflow depicted in both figure 2 and figure 5.

Comment 6: Figure References: In the text, refer explicitly to each figure when relevant steps are discussed.

Response 6: thank you, we have add figure 3-5 references in the text as suggested;

Comment 7: Minor Comments

Terminology Consistency: Ensure consistent use of terms like "XF Analyzer" vs. "Seahorse XF Pro Analyzer."

Response 7: thank you, we have used “Seahorse XF Pro Analyzer” in all the text, as suggested;

Comment 8: Typos:

Line 9: "metab-olism" should be corrected to "metabolism."

Line 15: "sus-pension" should be corrected to "suspension."

Response 8: thank you for pointing this out, as suggested, we have corrected the typos on lines 9 and 15;

Comment 9: Abbreviations: Provide a brief list of abbreviations (e.g., OCR, ECAR) at the beginning or end for clarity.

Response 9: thank you for pointing this out, as suggested, we have added the abbreviations list at lines 442-447.

Reviewer 2 Report

Comments and Suggestions for Authors

The topic of this protocol paper is of interest, as measuring metabolic activity in cancer environment is completely necessary and more techniques and protocols for this are required.  In this review, the authors explain point by point two protocols using Agilent Seahorse Technology to measure metabolic activity in suspension and adherent cancer cells. It is very well explained. I have some minor comments:

1) In the introduction, can you describe one or two more alternatives to the two methods that you describe.

2) Line 166, the size of the letter is smaller than the rest

 3) Even if every step is well described, please specify the errors/problems the users can find and how to solve them

 4) In the discussion, please address more deeply the limitations of the two methods described.

Author Response

Comment 1: The topic of this protocol paper is of interest, as measuring metabolic activity in cancer environment is completely necessary and more techniques and protocols for this are required.  In this review, the authors explain point by point two protocols using Agilent Seahorse Technology to measure metabolic activity in suspension and adherent cancer cells. It is very well explained. I have some minor comments:

Response 1: Thank you so much for your appreciation and your valuable advice and suggestions, which have greatly improved our Protocol;

Comment 2:  In the introduction, can you describe one or two more alternatives to the two methods that you describe.

Response 2: thank you for pointing this out, as suggested we have added: “The use of Oroboros O2K and Agilent Seahorse technologies for studying metabolism is expanding in this field. Oroboros O2K (Oroboros Instruments, Innsbruck, Austria) is a high-resolution respirometer (HRR) for measuring rates of mitochondrial respiration. However, the Oroboros O2K has some limitations. It only has two chambers, which makes high-throughput testing impractical without multiple O2K units. Additionally, the clean-ing protocols after each experiment are critical and procedure-specific, requiring up to three hours”, lines 62-68;

Comment 3: Line 166, the size of the letter is smaller than the rest

Response 3: thank you, we have adapted the font to the entire text;

Comment 4: Even if every step is well described, please specify the errors/problems the users can find and how to solve them

Response 4: thank you for pointing this out, as suggested, we have added some technical tips, errors and how to solve them;

Comment 5: In the discussion, please address more deeply the limitations of the two methods described

Response 5: thank you for pointing this out, as suggested, in the discussion on lines 467-477, 492-502, we have added more detail (limitations and advantages) of the two methods described.